# Effects of Non-Ionic Micelles on the Acid-Base Equilibria of a Weak Polyelectrolyte

**DOI:** 10.3390/polym14091926

**Published:** 2022-05-09

**Authors:** Evgenee Yekymov, David Attia, Yael Levi-Kalisman, Ronit Bitton, Rachel Yerushalmi-Rozen

**Affiliations:** 1Department of Chemical Engineering, Ben-Gurion University of the Negev, Beer-Sheva 84105, Israel; ekymov@post.bgu.ac.il (E.Y.); attiad@post.bgu.ac.il (D.A.); rbitton@bgu.ac.il (R.B.); 2The Center for Nanoscience and Nanotechnology, The Hebrew University of Jerusalem, Jerusalem 91904, Israel; yael.kalisman@mail.huji.ac.il; 3The Institute of Life Sciences, The Hebrew University of Jerusalem, Jerusalem 91904, Israel; 4The Ilse Katz Institute for Nanoscience and Technology, Ben-Gurion University of the Negev, Beer-Sheva 84105, Israel

**Keywords:** polyacrylic acid, PAA, weak polyelectrolytes, WPEs, acid base, equilibrium, pKa, pH, excluded volume, surfactant, Brij-S20, F108, F127

## Abstract

Weak polyelectrolytes (WPEs) are widely used as pH-responsive materials, pH modulators and charge regulators in biomedical and technological applications that involve multi-component fluid environments. In these complex fluids, coupling between (often weak) interactions induced by micelles, nanoparticles and molecular aggregates modify the pKa as compared to that measured in single component solutions. Here we investigated the effect of coupling between hydrogen bonding and excluded volume interactions on the titration curves and pKa of polyacrylic acid (PAA) in solutions comprising PEO-based micelles (Pluronics and Brij-S20) of different size and volume fraction. Titration experiments of dilute, salt-free solutions of PAA (5 kDa, 30 kDa and 100 kDa) at low degree of polymer ionization (α < 0.25) drive spatial re-organization of the system, reduce the degree of ionization and consequentially increase the pKa by up to ~0.7 units. These findings indicate that the actual degree of ionization of WPEs measured in complex fluids is significantly lower (at a given pH) than that measured in single-component solutions.

## 1. Introduction

Weak polyelectrolytes (WPEs) are polymeric molecules comprising covalently linked monomers of a weak acid (or base). Weak monomeric acids (bases) protonate or deprotonate according to the reversible chemical reactions: $\mathrm{HA}\;⇄{\;H}^{+}+A^{-}$. The degree of ionization α, of simple acid in ideal bulk solution, where all the reacting species are mobile is [1]:(1)$$\;\alpha =\frac{\left[A^{-}\right]}{\left[\mathrm{AH}\right]+\left[A^{-}\right]}=\frac{1}{1+10^{{\mathrm{pK}}_{0}-\mathrm{pH}}}$$

pH is the bulk pH and ${\mathrm{pK}}_{0}$ is the equilibrium constant of the dissociation reaction. This treatment assumes that there are no additional, coupled interactions in the system. In real systems the degree of ionization is usually characterized by an effective equilibrium constant defined as the pH in which half of the acid groups are protonated [2]. The ${\mathrm{pK}}_{\mathrm{eff}}$ is calculated via the Henderson–Hasselbalch equation [3] for a monomeric acid (base)
(2)$${\mathrm{pK}}_{\mathrm{eff}}=\mathrm{pH}+\log\left(\frac{1-\alpha }{\alpha }\right)$$

Unlike monomeric acids, in WPEs the acid-base balance is not only determined by the bulk pH, ionic strength and the dielectric constant of the solvent, but also by the polymer’s ability to regulate the charging of the chain (charge regulation, CR) and modify the local environment according to the complex interplay between electrostatic interactions, the conformational degrees of freedom and excluded volume interactions [4]. The complex solution behavior of WPEs was investigated thoroughly by theory, simulations and experiments (for recent reviews see [5,6,7,8]).

Early studies by Katchalsky, Gillis and co-workers [9,10] described the conformation-dependent free energy cost of ionization of an individual, flexible polyelectrolyte chain in a dilute, salt-free aqueous solution. They suggested that the connectivity of the chain suppresses the ionization and modifies the titration curve of weak polyelectrolytes as compared to a solution of monomers of similar composition, as described in the following equation:(3)$$\;\mathrm{pH}={\mathrm{pK}}_{0}-\log\left(\frac{1-\alpha }{\alpha }\right)+0.4343\times 2\lambda {\left(\frac{3}{\lambda \text{’}\mathrm{sj}}\right)}^{\frac{1}{3}}{\left(\frac{\varepsilon^{2}}{{\mathrm{Dk}}_{b}\mathrm{Tb}}\right)}^{\frac{2}{3}}\alpha^{\frac{1}{3}}$$
where ${\mathrm{pK}}_{0}$ is $-\log_{10}\left(\mathrm{Ka}\right)$ of the monomers, s is the number of monomers in the chain, b is the length of the statistical unit, j is the number of monomers per acid group, ε is the electron charge, D is the solvent dielectric constant, $k_{b}T$ is the Boltzmann constant and absolute temperature. λ and λ’ are parameters that depend on the extension of the coil, known to change slowly during the titration, and thus are assumed to be constant [11]. The simplied equation is often used to fit experimental data:(4)$$\mathrm{pH}={\mathrm{pK}}_{0}-\log\left(\frac{1-\alpha }{\alpha }\right)+\mathrm{const}{\left(\alpha \right)}^{\frac{1}{3}}.\;$$

Studies of polyacrylic acid and other WPEs in dilute solutions have shown the expected linear dependence of pH on $\alpha^{\frac{1}{3}}$ [8,9,11,12].

Later studies extended the treatment to semi-dilute and concentrated solutions [13], star polymers [14], grafted WPE brushes [15,16,17,18] and crosslinked networks [19,20,21].

WPEs are used as pH-responsive materials in a variety of biomedical and technological applications [22,23] including pH-controlled capture and release of drugs [24,25] and other functional molecules, responsive gels [26,27] and polymer-based artificial muscles [28]. In these applications the fluid environment is multi-component and comprises mixtures of molecular aggregates, micelles, liposomes and nanoparticles, where multiple (often weak) intermolecular interactions are active. In such complex fluids, it is often observed that the response of the WPE to the environmental pH deviates from that predicted from the equilibrium constant measured in single component solutions of the polyelectrolytes [29], suggesting that coupling between weak interactions may lead to significant deviations of the acid-base equilibria of WPEs in single component solutions.

Here we investigated the effect of coupling between hydrogen bonding and excluded volume interactions on the titration curves and pKa (-log Ka, the equilibrium constant of the acid-base titration) of polyacrylic acid (PAA) in solutions containing PEO-based micelles of different size and volume fraction. It is well known that at low pH (pH ≤ 3.25) highly protonated acrylic acid (proton donor) forms a hydrogen-bonded complex with the etheric oxygen (proton acceptor) of PEO (complex coacervation) [30,31,32,33]. The interaction is modulated by the degree of ionization of the carboxylic group, and decays at pH above the pKa of the acrylic acid [30,31,32,33]. 

Titration experiments of dilute, salt-free solutions of PAA (5 kDa, 30kDa and 100kDa) were carried out in micellar solutions of self-assembling block-copolymers F127 (PEO_100_-PPO_70_-PEO_100_), F108 (PEO_132_-PPO_50_-PEO_132_) with typical micelles diameter of 20-25 nm and Brij-S20 (Polyethylene glycol octadecyl ether) with typical micelles diameter of 5–6 nm (see Table 1).

The results indicate that the combination of hydrogen bonding and excluded volume interactions induced by the presence of nanometric, non-charged micelles may modify the shape of the titration curves and increase the pKa values of PAA by up to 0.7 pH units. Thus, at a given pH, PAA is significantly less ionized than what would be expected from the pKa values determined in aqueous solutions. 

These findings are relevant to most applications of WPEs that rely on their utilization in synthetic or bio-related complex fluids that contain micelles, vesicles and nanoparticles.

## 2. Experimental

### 2.1. Materials

#### 2.1.1. Polyelectrolytes

Propionic acid (PAc) ~ 99.5 wt% was purchased from Sigma Aldrich (Burlington, MA, USA). The reported pK_a_ is 4.87–4.9 [34,35,36]. PAc was used in the study, as the chemical structure of the acid is similar to that of the monomers of PAA (Figure 1).

Poly (acrylic acid), PAA of different molecular weights (Table 1) was purchased as aqueous solutions of 5 kDa (50 wt%) Polyscience Inc. (Niles, IL, USA), 30 kDa (30 wt%) and 100 kDa (35 wt%) (Sigma Aldrich, USA). The Mw, the number of Khun segments (N), hydrodynamic radius (R_H_), the measured overlap concentration (C*) (see the Appendix A for details) and the (measured) pKa of PAA are presented in Table 1.

**Table 1 polymers-14-01926-t001:** Polyacrylic acid properties in salt-free solutions [37,38,39].

Mw (kDa)	N	R_H_ in Water (nm)	C* (wt%)	pKa (α = 0.5)
5	54	1.6	*	6.0
30	314	4.7	4.51 ± 0.05	6.1
100	1047	9.6	2.31 ± 0.05	6.3

* below the entanglement length.

#### 2.1.2. Acid Base

HCl standard solution of 1N and 0.1N (Sigma Aldrich) and NaOH powder (Bio-Lab Ltd., Jerusalem, Israel) were used following calibration. 

#### 2.1.3. Non-Ionic Surfactant and Block-Copolymers

Brij-S20 Polyoxyethylene (20) stearyl ether was purchased from Sigma Aldrich (CAS# 9005-00-9, PB0109) (See Table 2).

Pluronics poly (ethylene glycol)-block-poly(propylene glycol)-block-poly(ethylene glycol) polymers, F108 PEO_100_-PPO_70_-PEO_100_ and F127 PEO_132_-PPO_50_-PEO_132_ were received as a gift from BASF (F108 product no. 583 106, F108 product no. 583062) and used as received (See Table 2).

#### 2.1.4. Solvents

Deionized water (DIW) (Millipore, 18 MΩ∙cm) and ethylene glycol (EG) > 99.8 wt%, (Sigma Aldrich, St. Louis, MI, USA) were used.

### 2.2. Preparation of Solutions

Aqueous solutions of Brij-S20 were prepared by dissolving the powder in DIW at room temperature. Solutions of F108 and F127 in DIW were prepared by stirring the polymer powder in DIW for 24 h at 0 °C for complete dissolution [46]. The solutions were stored at 4 °C for several days before further usage, followed by equilibration at 22 ± 1 °C. 

NaOH solutions were prepared by dissolving NaOH powder in DIW followed by volumetric calibration by HCl 1N standard with phenolphthalein indicator. The obtained solution was diluted to the final concentration and calibrated again.

PAA solutions were prepared by mixing of PAA stock solution with DIW or the relevant solutions to a final concentration. 

### 2.3. Characterization

#### 2.3.1. Automatic Potentiometric Titration

Titrations were performed at 22 ± 1 °C using a homemade autotitrator based on Orion star a214 pH meter equipped with METER TOLEDO glass pH electrode, with ceramic junction ARGENTHAL™ Ag+-trap (Ag/AgCl) and 3 mol/L KCl reference electrolyte. The pH meter was calibrated by standard buffers with pH = 4.01, 7 and 10.04. Automatic titration was carried out using a weight-calibrated syringe pump (NE-1000 Pump, New Era Pump Systems Inc, Farmingdale, New York) and controlled by a computer.

The following sequence was used in the measurements: The pH of unstirred solution (15 mL) was measured, 20–35 μL titrant (NaOH or HCl) was injected and stirred for 45 s, allowed to rest for 90 s and re-measured.

#### 2.3.2. FTIR-ATR Measurement

Absorption measurements were carried out using a Thermo Scientific Nicolet iS50R spectrometer, Thermo Scientific, Waltham, Massachusetts. Thermo SpectraTech ARK 45 deg ZnSe crystals were used in a Pike Technologies VeeMAX III variable angle accessory. The spectrometer and Attenuated Total refractometry (ATR) accessory were continuously purged with 99.999% N_2_ during measurements. Liquid nitrogen cooled MCT-A detector was used to collect and average 128 scans at a resolution of 4 cm^−1^.

ZnSe crystal was used for background measurements in water for all the samples. The absorption peaks of water vapor and carbon dioxide were subtracted from the spectra using Ominic 9 software.

The carboxylate group is characterized by absorption bands in the 1000–1800 cm^−1^ region of the infrared spectrum [47]. Appendix A presents the characteristic frequencies at pH 2 (Appendix A), where the carboxylate groups are completely protonated. The characteristic frequencies are C=O at 1717 cm^−^^1^, CH_2_ stretching at 1455 cm^−1^ and the C-O stretch at 1265 cm^−1^. At pH 13 (Appendix A), the functional groups are fully ionized, and the characteristic frequencies are the symmetric and antisymmetric stretching of the carboxylate ion (COO-) at 1408 and 1562 cm^−1^.

The degree of ionization of PAA (and PAc) was calculated by linear superposition of the intensities of the absorption spectra of protonated and charged COO- units at a known pH values following the equation (see Appendix A): (5)$$A=A_{\mathrm{acid}}*\left(1-\alpha \right)-A_{\mathrm{base}}*\alpha$$

A-absorption of acid solution at known pH. $A_{\mathrm{acid}\backslash \mathrm{base}}$ is the absorption of the protonated (pH = 2) and fully ionized (pH = 13) acid groups. 

#### 2.3.3. Surface Tension

The surface tension of the different solutions was measured via the Du Nouy ring method (KRUSS digital tensiometer K10T), at 22 ± 1 °C.

#### 2.3.4. Small-Angle X-ray Scattering (SAXS)

Scattering patterns of the solutions were collected using SAXSLAB GANESHA 300-XL Xenocs, Grenoble, France. CuΚα radiation was generated by a Genix 3D Cu-source with an integrated monochromator, 3-pinhole collimation and a two-dimensional Pilatus 300K detector. The scattering intensity I(q) was recorded in the interval of 0.007 < q < 0.25 Å^−1^ (corresponding to the length scale of 25–900 Å), where the scattering vector is defined as $q=\left(4\pi /\lambda \right)\cdot \sin\theta$, with 2θ and λ being the scattering angle and wavelength, respectively. The measurements were performed under vacuum at an ambient temperature (~22 °C). The solutions were sealed in thin-walled quartz capillaries about 1.5 mm in diameter and 0.01 mm wall thickness; the scattering curves were corrected for counting time and sample absorption. The 2D SAXS patterns were azimuthally averaged to produce one-dimensional intensity profiles, I vs. q, using the two-dimensional data reduction program SAXSGUI. The scattering spectra of the solvent were subtracted from the corresponding solution data using the Irena package [48] in Igor Pro 9 from WaveMetrics (Portland, Oregon), for analysis of small angle scattering data. Data analysis was based on fitting the scattering curve to an appropriate model using the SasView program version 4.2.2, University of Tennessee, Knoxville, TN, USA [49].

#### 2.3.5. Transmission Electron Microscopy (TEM)

Cryo-TEM: Rapid cooling enables direct imaging of molecular assemblies and nanostructures in aqueous media. The samples were prepared by applying a 3 μL drop to a TEM grid (300 mesh Cu Lacey substrate, Ted Pella, Ltd., Redding, CA, USA) following a short pre-treatment of the grid via glow discharge. The excess liquid was blotted and the specimen was vitrified by rapid plunging into liquid ethane precooled by liquid nitrogen using a vitrification robot system (Vitrobot mark IV, FEI, Thermo-Fisher Scientific, Waltham, MA, USA). The rapid cooling results in physical fixation of the liquid state, so as to preserve the native structures. Thus, it allows examination of the polymeric assemblies in the high vacuum of the electron microscope at cryogenic temperature, which prevents the formation of either cubic- or hexagonal ice. The vitrified samples were examined at −177 °C using FEI Tecnai 12 G^2^ TWIN TEM operated at 120 kV and equipped with a Gatan model 626 cold stage. The images were recorded by a 4K×4K FEI Eagle CCD camera in low dose mode. TIA (Tecnai Imaging & Analysis, Thermo-Fisher Scientific, Waltham, MA, USA) software was used to record the images.

Negative staining: To increase the inherently low contrast of the polymer assemblies, transmission electron microscopy of dried, stained samples was performed. The solution (2 μL) was applied to a glow discharged TEM grid (carbon supported film on 300 mesh Cu grids, Ted Pella, Ltd., Redding, CA, USA). The excess liquid was blotted, and the grids were washed on two droplets of de-ionized water following by staining with 2% uranyl acetate for 40 s. The grids were blotted and dried under ambient conditions at room temperature before they were observed by FEI Tecnai 12 G^2^ TWIN TEM operated at 120 kV. The images were recorded by a 4K×4K FEI Eagle CCD camera and using TIA software (Tecnai Imaging and Analysis. Technai 3.0 FEI Hillsboro, Oregon, USA).

#### 2.3.6. Viscosity Measurements

The overlap concentration, C*, of PAA 30 kDa, and 100 kDa were calculated from the measured intrinsic viscosity of solutions at concentrations of 0.002–0.015 mg/mL. The method is based on the measurement of the difference in flow time via a glass capillary (5 cm length and 50 mm diameter, Ubbelohde tube) due to viscosity differences between pure solvent (water) and polymer solution (in the dilute state), at 22 ± 1 °C, maintained using a thermal bath.

The inherent and reduced viscosity was calculated from the following relations: (6)$${\;\eta }_{\mathrm{inh}}=\ln\left(\frac{t}{t_{0}C}\right)$$
(7)$$\eta_{\mathrm{red}}=\frac{t-t_{0}}{t_{0}C}$$
where $\eta_{\mathrm{inh}}$ is the inherent viscosity, t is the polymer solution flow time at a specific concentration, $t_{0}$ is the pure solvent flow time, C is the polymer concentration and $\eta_{\mathrm{red}}$ is the reduced viscosity.

The extrapolation of the inherent and reduced viscosity to zero concentration provides the intrinsic viscosity, $\eta_{\mathrm{int}}$, and the overlap concentration, C* is obtained from:(8)$$\eta_{\mathrm{int}}=\frac{1}{C^{*}}$$

The measured values of C* are presented in Table 1 (and Appendix A).

## 3. Results

Potentiometric titrations of PAA of different molecular weights were carried out in salt-free, dilute (0.1 wt% and 1 wt%, see Table 1) and in micellar solutions of Brij-S20, F127 and F108 as described below. Direct measurements of the degree of ionization of PAA (α) as a function of the pH of the solution in native and micellar solutions carried out via FTIR in the ATR mode, resulted in similar titration curves to those measured via potentiometry (Appendix A). Thus, most of the measurements were performed using potentiometric-based automatic titration.

### 3.1. Titration Experiments

Titration curves of PAA in dilute, salt-free solutions were used to reiterate the expected deviation of the polyacids from the ideal Henderson–Hasselbalch curve (Equation (1)) in micelle-free solutions. In these experiments propionic acid (PAc, Figure 1 black curve) served as the monomeric reference to PAA, as the chemical structure of PAc is similar to that of the PAA monomer (see Figure 1).

PAA polymers in aqueous solutions (Figure 1A,B) are observed to exhibit the behavior predicted by the Katchalsky and Gillis model (Equation (4)): The titration curves of PAA (5, 30 and 100 kDa) show the typical lower degree of ionization for a given bulk pH (the shift of the titration curve to higher pH), as compared to propionic acid (PAc), and the broadening of the curve where the electrostatic potential is pH dependent [50]. At low concentrations the measured degree of ionization decreases with the molecular weight. The effect is less pronounced at 1 wt% PAA. The titration curves presented in Figure 1C,D are centered around the pH corresponding to α = 0.5 for better visualization of the deviations of the titration curves of PAA from that of propionic acid.

Titration curves of PAA in micellar solutions are presented in Figure 2. The mixtures contain aqueous solutions of PAA (0.1 wt% or 1 wt%, volume fraction ϕ = 1.2 × 10^−3^ or 1.2 × 10^−2^, respectively) + Brij-S20 1 wt% (volume fraction, ϕ = 9 × 10^−3^) or F127 5 wt%, (volume fraction ϕ 4.5 × 10^−2^). The micelles are observed in TEM images (Appendix A). In the micellar solutions the titration curves become narrower, and for a given pH (for pH < pH_(α = 0.5)_), the degree of ionization is lower than that of the corresponding PAA in water. As expected, the effect is more pronounced at low PAA concentrations and high molecular weights (see the discussion, Figure 6).

Titration curves of PAc (Appendix A) show only constant shift of the pKa of up to ~0.1 pH units, and the shape of the titration curve is not modified. As the specific interactions (here, hydrogen bonding) between the carboxylic groups of the acids and the etheric groups of the micelles are similar in PAc and PAA, the results indicate that the polymeric nature of PAA is the origin of the observed deviation.

The effect of F127 and F108 concentration on the reduction in the degree of ionization of PAA as compared to Pluronic-free solutions was investigated in a series of titrations (Appendix A). The calculated pKa as a function of α^1/3^ is presented in Figure 3. In solutions of F108, the pKa of PAA is significantly increased relative to that of PAA in water at F108 concentration above the CMC (3 wt%, Table 2 and Appendix A): While F108 molecules (bright green line in Figure 3A) induce a deviation of about 0.15 of the PAA 30kDa pKa, F108 micelles (olive green) increase the deviation to about 0.52 units. In micellar solutions of F127 (concentrations of 3, 5 and 7 wt%, all above the CMC, Table 2) we observe that the dependence of the deviation increases as the concentration of F127 increases from 3 wt% to 5 wt%, reaching a value of ΔpKa = 0.76 but levels-off, as the F127 concentration is increased from 5 wt% to 7 wt% (the azure and blue curves in Figure 3B,D).

Very differently, the value of the pKa of propionic acid in the presence of micelles is almost constant with α^1/3^ (dashed black curve in Figure 3A,C).

The sensitivity of the deviations to the concentration of F108 is less pronounced at higher PAA concentrations (Appendix A).

### 3.2. PAA–Micellar Interactions

The measurements presented so far indicate that non-charged micelles modify the acid-base equilibria and the resulting titration curves of PAA. To test the origins of the effect we characterized the impact of solvated PAA (and PAc) on the surface activity and self-assembly of Pluronics (and Brij-S20) at pH = 3.5 and 7. The modification of the CMC of Pluronics block-copolymers was used before as a measure of the strength of the interaction between Pluronics and ionic surfactants [51,52].

Surface tension measurements of mixtures of F127 (Figure 4A), or Brij-S20 (Figure 4B) with PAA solutions (30 kDa, concentration of 0.1 wt%) exhibit similar curves at the two pH values measured here. In particular, the assembly of F127 (and Brij S-20), indicated by the breaks in the curve, occurs at similar concentration at both pH. In liquid mixtures of PAA + F127 (or Brij-S20) we observe that the first break in the curve (indicative of Pluronics self-assembly) is similar to that measured in the native solution (0.030 ± 0.002 wt%). The surface access calculated from the Gibbs adsorption isotherm [53], in the native solution of F127 is 1.7 × 10^−3^ mol/m^2^ and 2.5 × 10^−3^ mol/m^2^ in the F127-PAA solution, at concentrations below the break in the curve. At concentrations above the break in the curve the surface access of 5.7 × 10^−4^ mol/m^2^ in F127 solutions is reduced to almost zero in the F127-PAA mixtures. These differences in the CMC and the surface accesses are minor as compared with those reported in mixtures of F127 and ionic surfactants [51], indicating weak intermolecular interactions between PAA and F127, and a minor effect on the formation of F127 micelles. A qualitatively similar behavior is observed in solutions of Brij-S20, (Figure 4B), while the overall reduction in the surface tension due to the presence of PAA is higher. The qualitatively similar behavior of F108-PAA mixtures is presented in the Appendix A and Appendix A). 

We note here that the titration experiments presented in this study were carried out at concentrations well above the CMC for F127 and Brij-S20, and for F108 at concentrations both below and above the CMC (Figure 3). At these concentration regimes the effect of PAA on the spatial organization of the micelles was also investigated using SAXS. 

### 3.3. Characterization of PAA–Micelle Solutions Using SAXS

SAXS measurements of aqueous mixtures of PAA (1 wt%) and micellar solutions of F127 (5 wt%) at pH = 3.5 are presented in Figure 5A. For comparison, scattering patterns of F127 in water (curve 1) and in solution of 1 wt% PAc (curve 2) are presented as well. The curves show that in the presence of solvated PAA (30 kDa) (Figure 5A, Curve 3) the peak at q ~ 0.07 A^−1^, characteristic of the form factor of the F127 micelles, is more pronounced than that in water [54]. The curve is well fitted by a core–shell–sphere form factor, with Schulz distribution [54] of the core radius to account for the polydispersity (Equations (S4) and (S5) and Appendix A). An additional peak at q ~0.03 A^−1^ indicates inter-micellar correlations that can be fitted using a hard-sphere structure factor (described in detail in the Appendix A). In these PAA-F127 mixtures the calculated volume fraction of F127 micelles in solutions of 5 wt% is found to increase from ϕ ~0.084 to ϕ ~0.187 (Appendix A). These observations are consistent with a higher local concentration of the F127 micelles, as would be observed if the interaction with PAA (at pH = 3.5) induces crowding of the micelles [55]. Note that a similar effect is not observed in the presence of 1 wt% PAc (Curve 2).

The scattering curves obtained from similar mixtures at pH = 7 are presented in Figure 5B: Scattering from F127 solutions (Curve 1), mixtures of F127-PAc (Curve 2) and F127-PAA (Curve 4). While the scattering patterns of F127 and F127-PAc can be well fitted to a core–shell–sphere form factor combined with a hard-sphere structure factor (Appendix A), characteristic of micellar solutions of F127, the scattering curve of F127-PAA cannot. However, it appears that in the higher q range curve 4 resembles curve 3 which presents the scattering curve obtained from solvated PAA (30 kDa, 1 wt% solution). This curve exhibits a correlation peak at q_0_ ~0.0602 A^−1^ corresponding to d_0_ = 2π/q_0_ (the average distance between the charged PAA segments) characteristic of a WPE in a salt-free solution, below C* [56]. 

To evaluate the role of the interactions between PAA and F127 in the spatial organization of the F127 micelles, we performed superposition of the SAXS curves measured in single-component solutions (aqueous solution of F127 and aqueous solution of PAA), see Figure 5C for pH = 3.5 and Figure 5D for pH = 7. As observed in Figure 5C, at pH = 3.5 the scattering pattern measured from solutions of F127+PAA (blue curve) cannot be re-constructed by superposition of the curves measured from single-component solutions (orange curve), indicating that the mutual interactions are the origin of the spatial re-organization of the F127 micelles. At pH = 7 (Figure 5D) the calculated (orange curve) and measured (blue curve) scattering patterns are similar, indicating that PAA-F127 interactions do not modify significantly the spatial organization of the micelles. 

Similar analysis carried out in mixtures of PAA with Brij-S20 or with F108 (presented in Appendix A) shows similar dependence on the pH, with micelle–WPE interactions leading to spatial re-organization at pH = 3.5 and lack of spatial effects at pH = 7. The curves of PAA at pH = 3.5 and 7 were fitted to the Debye Gaussian coil model [57] and to the broad Lorentzian peak model (Equations (S2) and (S3) of the Appendix A), respectively (Appendix A).

TEM images of dried, negatively stained samples prepared from solutions of F127 micelles in PAA solutions presented in Figure 5E,F, show aggregated, not well-defined micelles at pH = 3.5 (Figure 5E), and non-aggregated, more uniform, micelles at pH = 7 (Figure 5F). The dimensions of the micelles are consistent with those obtained from the SAXS analysis (Appendix A).

## 4. Discussion

The acid-base equilibria of WPEs have been investigated thoroughly over the last 60 years [9,13]. It is well established that the dissociation of polyacids is suppressed as compared to the corresponding monomeric acids. For example, while the degree of ionization of Propionic acid at pH = 6 (~pKa+1) is 90%, the degree of ionization of PAA of low molecular weight at this pH is about 40% and that of high molecular weight is about 60% (Figure 1). Furthermore, for WPEs of a given molecular weight and concentration, the pKa is a parameter that depends on the degree of ionization (as indicated by Equations (3) and (4)) and thus a single value of pKa cannot be used to characterize the shift of the acid-base equilibrium throughout the titration curve.

The deviations of WPEs from the behavior of monomeric acids can be interpreted as manifestations of Le Chatelier’s principle: In WPEs the dissociation of a covalently linked acid group is correlated, via the conformational change of the chain to the dissociation of any other acid group along the chain [58]. In addition, it is well established that the conformation dependent free energy of ionization in WPEs is sensitive, in salt-free solutions to the solvent and to the weak interactions (as compared to the thermal energy) with neighboring chains [13]. Yet, there is still a gap in characterization of the effect of non-charged molecular assemblies, such as micelles, soft and hard nanoparticles and fibers that are abundant in the complex fluids environment relevant for most of the applications of WPEs.

Here we investigated the effect of non-charged micelles comprising PEO-PPO units on the acid-base response of PAA. First, titrations experiments, in salt-free solutions of PAA with molecular weights of 5 kDa, 30 kDa and 100 kDa (at volume fraction ϕ = 0.0012 and ϕ = 0.012) were carried out via potentiometric methods (and complemented by direct measurement of the degree of ionization via FTIR-ATR, Appendix A). The titration curves exhibit the expected deviations from the titration curves of monomeric acids of similar composition (PAc, Figure 1) and reveal the expected dependence of the pKa on the molecular weight (Figure 1). We note that in the conditions investigated here the ionic strength was below 15 mM (at α < 0.25) and thus can be neglected.

These were followed by titration experiments of PAA in solutions of Brij-S20 (ϕ = 0.009), F127 and F108 ϕ = 0.045) at different concentrations.

The main results of the study are summarized in Figure 6, where the deviation of the pKa (ΔpKa) measured in micellar solutions from that measured in surfactant-free aqueous solutions are presented. For example, the difference between the degree of dissociation of PAA 100 kDa (0.1 wt%, α = 0.1) in water and in micellar solutions (1 wt% Brij-S20) is >0.6 (Figure 6 A). A similar deviation is observed for PAA 30 kDa, and the effect is smaller for the shorter PAA (5 kDa, Figure 6A). A similar behavior is observed for micelles of F108 and F127. In addition, it is found that the effect of micelles on the acid-base equilibria of PAA depends on the pH: At pH ~4.6, for example, (Figure 6A) the presence of micelles of typical dimensions similar to that of the polymers (diameter of 3-6 nm for Brij-S20 and 20-25 nm, for F127 and F108) significantly modifies the degree of dissociation of PAA, whereas the effect is much smaller at pH ~6.4 (Figure 6C). The deviations from the Katchalsky and Gillis model induced by the presence of the micelles are presented in Figure 6D,E: In water the pKa values of PAA (black line) show the reported dependence on the degree of ionization [12], while in micellar solutions a non-monotonous deviation from that of PAA in water is observed. At a higher PAA concentration (Figure 6E and the Appendix A) the deviations are reduced. The effect is significantly smaller for solutions of F108 chains, at concentrations below the CMC (3 wt% of F108) and for micellar solutions of F127 which saturates as the micelle concentration increases (3 to 7 wt% F127) (Figure 3). It is important to note that only a minor, constant deviation of about ~0.1 pH units is observed in titrations of PAc (Appendix A) in micellar solutions, and the shape of the curve is not modified.

The observed pKa shift of 0.7 units relates to a free energy cost of ~4 KJ/mol, while the hydrogen bonding between hydrogenated PAA and PEO is of the order of ~0.57 KJ/mol [60]. Therefore, hydrogen bonding alone could not suffice to induce the observed effect in Δpka.

In addition, surface tension measurements presented in Figure 4 show only a minor effect on the surface activity and the CMC of the Pluronics block-copolymers (and Brij-S20). 

Furthermore, complexation of the micelles with PAc, a monomeric acid of similar composition and structure to that of PAA monomers (Figure 1) does not significantly shift the pKa of the acid (Figure 3A,C and Appendix A). SAXS measurements (Figure 5) reveal that the PAA chains, but not the monomeric PAc acid, induce crowding of the attached PEO-based micelles at low pH: At concentrations of F127 (and Brij-S20) more than twofold above the CMC, at pH = 3.5, the spatial organization of the micelles is significantly modified by the interaction with PAA (Figure 5). At this pH, the interaction with PAA leads to local crowding of the micelles (Figure 5A,C,E). At pH = 7 such an effect is not observed, and the presence of PAA does not affect the shape, dimensions or the local organization of the micelles (Figure 5B,D,F).

The observations presented in this study indicate that the necessary conditions for modification of the acid-base equilibrium of PAA and the resulting pKa are three-fold: Hydrogen bonding [60], the presence of micelles at relatively high volume fraction and long-enough-PAA chains. All three elements seem to be necessary for shifting the acid-base equilibria of the PAA, probably by modifying the conformational space of the PAA due to excluded volume interactions.

A schematic overview of the interactions in the micellar solutions is presented in Figure 7.

We can estimate the relative importance of the observed shift in the degree of ionization by comparing it to deviations from the acid-base equilibria of star-like PAA [61] where polymer topology modifies significantly the conformational space of the chains. Experimental studies of the acid-base equilibria of randomly branched PAA reported a maximal deviation of about 1 pH unit [62], along with significant modification of the equilibrium and dynamic properties, as compared to linear PAA.

We thus suggest that the combination of relatively weak interactions, and generic excluded volume effects present in multi-component complex fluids environments, can significantly shift the acid-base equilibria of WPEs. Therefore, the degree of ionization of WPEs measured in multi-component fluids may differ significantly from that expected from pKa values obtained in single-component solutions.

## 5. Conclusions

The sensitivity of the acid-base equilibrium of WPEs to electrostatic effects, hydrogen bonding, dipolar and van-der-Walls interactions was thoroughly investigated in a variety of systems [4,63]. The role of chain connectivity and in particular the effect of conformational entropy modifications, via external constraints, on the acid-base equilibria of WPEs was investigated as well [14,15,16,17], though understanding of the full consequences of steric effects is still a challenge. 

While applications of WPEs rely on the ability to predict and control the acid-base equilibrium, the resulting pKa and the degree of ionization, it is often observed that when embedded in complex fluids, their pH-response differs significantly from that predicted by measurements carried out in single-component liquids.

Here we characterized the effect of non-ionic micelles on the titration curves (and the calculated pKa) of PAA, in salt-free, dilute solutions. Titration curves of PAA at different molecular weights in solutions of PEO-based micelles (Brij-S20, F108 and F127) revealed that the non-ionic micelles affect the acid-base equilibria of the PAA, and shift the pKa to higher values, while they have only a minor effect on the titration curves and pKa of propionic acid, at similar concentrations.

Combination of surface tension measurements, TEM study and SAXS investigation of the PAA–micellar solutions at low and high pH values reveals that weak interaction between the components is sufficient to induce crowding of the micelles at low pH, where coupling between specific interaction (hydrogen bonding) and excluded volume interactions (due to micelles crowding) shifts the pKa of PAA by up to 0.7 pH units. 

We suggest that coupling between weak interactions in fluids comprising different populations of molecular aggregates may modify the acid-base responsivity of WPEs.

## Data Availability

The data presented in this study are available on request from the corresponding author.

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
