# Peer review of "Effects of Non-Ionic Micelles on the Acid-Base Equilibria of a Weak Polyelectrolyte"

_polymers, 2022, doi:10.3390/polym14091926_

Round 1
Reviewer 1 Report
The authors reported their study on acid-base equilibria in a system of a weak polyelectrolytes (PAA) and non-ionic micelles formed by polymeric surfactants in aqueous solutions. Combined with the titration experiments and FT-IR measurements, the authors characterized the ionization degrees and pKa values of PAA at different pH with/without the surfactants. Furthermore, the authors combined SAXS and surface tension measurement, to investigate the interaction between PAA with the surfactants. Based on the above information, the authors rationalized the effect of polymeric surfactants on the ionization of PAA in aqueous solutions. The referee believes that the author studied a system which is fundamental and important for understanding polyelectrolytes in solutions. They provided solid characterizations of the systems of interest and decent discussion to support their conclusions. The referee believes this manuscript can be published, after addressing the following issues:
Material: Was PAA purified before use, e.g., dialysis to remove small molecular weight salts and impurities?
Equation 3: The coefficient before ? is “0.4343.2”. Is it correct?
Figure 2: How were the pKa values calculated? Were they based on the Equation 4?
Figure 3, S9, S10: What are the titration results of surfactants solutions (Pure water, Brij-S20, F108 and F127), without PAA polymers? As per the referee’s perspective, the surfactants itself may influence the pH of the solution. To separate the effect of surfactants-PAA polymer interactions on pH with the effect from the surfactants itself, it is necessary to check if the surfactants can change the titration curves, by comparing the titration curves of surfactant solutions with pure water.
Figure 6: In caption: “Shift of pKa values …… Similar plots are presented for PAA 5kDa and PAA 100 kDa in ESI Figures S13.” However, Figure S13 is a set of SAXS plots, not shifts of pKa values.
Besides, the referee is also curious about the Figure 5, PAA-F127 SAXS curve. It shows a downturn towards low q and a correlation peak at q ~ 0.03 A-1, which suggests strong interparticle interactions. However, this was not observed in other surfactants systems. Is there any possible reason for this?
Reviewer 2 Report
The manuscript authored by Yekymov et al. describes well-designed studies on the influence of Pluronics and Brij-20 micelles on the acid base equilibria of WPE.
The presented studies are very well planned and executed. The discussion is correct and scientifically sound.
Even though I could do some things differently, I do not think that such comments would enrich this work, because it is ready to be published as it is.
I have only one technical question for the authors:
-Line 160 concerning the liquid nitrogen cooled DTSG detector - I wonder why, because as far as I know the DTGS detector is FE cooled (Peltier pump), and the detector cooled with liquid nitrogen is MCT? Please explain this to satisfy my curiosity.
